# Dual Discriminator Generative Adversarial Nets

**Tu Dinh Nguyen, Trung Le, Hung Vu, Dinh Phung**
Deakin University, Geelong, Australia
Centre for Pattern Recognition and Data Analytics
{tu.nguyen, trung.l, hungv, dinh.phung}@deakin.edu.au

## Abstract

We propose in this paper a novel approach to tackle the problem of mode collapse encountered in generative adversarial network (GAN). Our idea is intuitive but proven to be very effective, especially in addressing some key limitations of GAN. In essence, it combines the Kullback-Leibler (KL) and reverse KL divergences into a unified objective function, thus it exploits the complementary statistical properties from these divergences to effectively diversify the estimated density in capturing multi-modes. We term our method *dual discriminator generative adversarial nets* (D2GAN) which, unlike GAN, has *two* discriminators; and together with a generator, it also has the analogy of a minimax game, wherein a discriminator rewards high scores for samples from data distribution whilst another discriminator, conversely, favoring data from the generator, and the generator produces data to fool both two discriminators. We develop theoretical analysis to show that, given the maximal discriminators, optimizing the generator of D2GAN reduces to minimizing both KL and reverse KL divergences between data distribution and the distribution induced from the data generated by the generator, hence effectively avoiding the mode collapsing problem. We conduct extensive experiments on synthetic and real-world large-scale datasets (MNIST, CIFAR-10, STL-10, ImageNet), where we have made our best effort to compare our D2GAN with the latest state-of-the-art GAN's variants in comprehensive qualitative and quantitative evaluations. The experimental results demonstrate the competitive and superior performance of our approach in generating good quality and diverse samples over baselines, and the capability of our method to scale up to ImageNet database.

## 1 Introduction

Generative models are a subarea of research that has been rapidly growing in recent years, and successfully applied in a wide range of modern real-world applications (e.g., see chapter 20 in [9]). Their common approach is to address the density estimation problem where one aims to learn a model distribution $p_{\text{model}}$ that approximates the true, but *unknown*, data distribution $p_{\text{data}}$. Methods in this approach deal with two fundamental problems. First, the learning behaviors and performance of generative models depend on the choice of objective functions to train them [29, 15]. The most widely-used objective, considered the de-facto standard one, is to follow the principle of maximum likelihood estimate that seeks model parameters to maximize the likelihood of training data. This is equivalent to minimizing the Kullback-Leibler (KL) divergence between data and model distributions: $D_{\text{KL}}(p_{\text{data}} \| p_{\text{model}})$. It has been observed that this minimization tends to result in $p_{\text{model}}$ that covers multiple modes of $p_{\text{data}}$, but may produce completely unseen and potentially undesirable samples [29]. By contrast, another approach is to swap the arguments and instead, minimize: $D_{\text{KL}}(p_{\text{model}} \| p_{\text{data}})$, which is usually referred to as the *reverse* KL divergence [23, 11, 15, 29]. It is observed that optimization towards the reverse KL divergence criteria mimics the mode-seeking process where $p_{\text{model}}$ concentrates on a *single* mode of $p_{\text{data}}$ while ignoring other modes, known as the problem of *mode collapse*. These behaviors are well-studied in [29, 15, 11].

The second problem is the choice of formulation for the density function of $p_{\text{model}}$ [9]. One might choose to define an *explicit* density function, and then straightforwardly follow maximum likelihood framework to estimate the parameters. Another idea is to estimate the data distribution using an *implicit* density function, without the need for analytical forms of $p_{\text{model}}$ (e.g., see [11] for further discussions). One of the most notably pioneered class of the latter is the generative adversarial network (GAN) [10], an expressive generative model that is capable of producing sharp and realistic images for natural scenes. Different from most generative models that maximize data likelihood or its lower bound, GAN takes a radical approach that simulates a game between two players: a generator $G$ that generates data by mapping samples from a noise space to the input space; and a discriminator $D$ that acts as a classifier to distinguish *real* samples of a dataset from *fake* samples produced by the generator $G$. Both $G$ and $D$ are parameterized via neural networks, thus this method can be categorized into the family of deep generative models or generative neural models [9].

The optimization of GAN formulates a minimax problem, wherein given an optimal $D$, the learning objective turns into finding $G$ that minimizes the Jensen-Shannon divergence (JSD): $D_{\text{JS}}\left(p_{\text{data}} \| p_{\text{model}}\right)$. The behavior of JSD minimization has been empirically proven to be more similar to reverse KL than to KL divergence [29, 15]. This, however, leads to the aforementioned issue of mode collapse, which is indeed a notorious failure of GAN [11] where the generator only produces similarly looking images, yielding a low entropy distribution with poor variety of samples.

Recent attempts have been made to solve the mode collapsing problem by improving the training of GAN. One idea is to use the minibatch discrimination trick [27] to allow the discriminator to detect samples that are unusually similar to other generated samples. Although this heuristics helps to generate visually appealing samples very quickly, it is computationally expensive, thus normally used in the last hidden layer of discriminator. Another approach is to unroll the optimization of discriminator by several steps to create a surrogate objective for the update of generator during training [20]. The third approach is to train many generators that discover different modes of the data [14]. Alternatively, around the same time, there are various attempts to employ autoencoders as regularizers or auxiliary losses to penalize missing modes [5, 31, 4, 30]. These models can avoid the mode collapsing problem to a certain extent, but at the cost of computational complexity with the exception of DFM in [31], rendering them *unscalable* up to ImageNet, a large-scale and challenging visual dataset.

Addressing these challenges, we propose a novel approach to both effectively avoid mode collapse and efficiently scale up to very large datasets (e.g., ImageNet). Our approach combines the KL and reverse KL divergences into a unified objective function, thus it exploits the complementary statistical properties from these divergences to effectively diversify the estimated density in capturing multi-modes. We materialize our idea using GAN's framework, resulting in a novel generative adversarial architecture containing three players: a discriminator $D_1$ that rewards high scores for data sampled from $p_{\text{data}}$ rather than generated from the generator distribution $p_G$ whilst another discriminator $D_2$, conversely, favoring data from $p_G$ rather $p_{\text{data}}$, and a generator $G$ that generates data to fool both two discriminators. We term our proposed model *dual discriminator generative adversarial network* (D2GAN).

It turns out that training D2GAN shares the same minimax problem as in GAN, which can be solved by alternatively updating the generator and discriminators. We provide theoretical analysis showing that, given $G$, $D_1$ and $D_2$ with enough capacity, i.e., in the nonparametric limit, at the optimal points, the training criterion indeed results in the minimal distance between data and model distribution with respect to both their KL and reverse KL divergences. This helps the model place fair distribution of probability mass across the modes of the data generating distribution, thus allowing one to recover the data distribution and generate diverse samples using the generator in a single shot. In addition, we further introduce hyperparameters to stabilize the learning and control the effect of each divergence.

We conduct extensive experiments on one synthetic dataset and four real-world large-scale datasets (MNIST, CIFAR10, STL-10, ImageNet) of very different nature. Since evaluating generative models is notoriously hard [29], we have made our best effort to adopt a number of evaluation metrics from literature to quantitatively compare our proposed model with the latest state-of-the-art baselines whenever possible. The experimental results reveal that our method is capable of improving the diversity while keeping good quality of generated samples. More importantly, our proposed model can be scaled up to train on the large-scale ImageNet database, obtain a competitive variety score and generate reasonably good quality images.

In short, our main contributions are: (i) a novel generative adversarial model that encourages the diversity of samples produced by the generator; (ii) a theoretical analysis to prove that our objective is optimized towards minimizing both KL and reverse KL divergence and has a global optimum where $p_G = p_{\text{data}}$; and (iii) a comprehensive evaluation on the effectiveness of our proposed method using a wide range of quantitative criteria on large-scale datasets.

## 2   Generative Adversarial Nets

We first review the generative adversarial network (GAN) that was introduced in [10] to formulate a game of two players: a discriminator $D$ and a generator $G$. The discriminator, $D(\mathbf{x})$, takes a point $\mathbf{x}$ in data space and computes the probability that $\mathbf{x}$ is sampled from data distribution $P_{\text{data}}$, rather than generated by the generator $G$. At the same time, the generator first maps a noise vector $\mathbf{z}$ drawn from a prior $P(\mathbf{z})$ to the data space, obtaining a sample $G(\mathbf{z})$ that resembles the training data, and then uses this sample to challenge the discriminator. The mapping $G(\mathbf{z})$ induces a generator distribution $P_G$ in data domain with probability density function $p_G(\mathbf{x})$. Both $G$ and $D$ are parameterized by neural networks (see Fig. 1a for an illustration) and learned by solving the following minimax optimization:

$$\min_G \max_D \mathcal{J}(G, D) = \mathbb{E}_{\mathbf{x} \sim P_{data}(\mathbf{x})}\left[\log\left(D(\mathbf{x})\right)\right] + \mathbb{E}_{\mathbf{z} \sim P_{\mathbf{z}}}\left[\log\left(1 - D(G(\mathbf{z}))\right)\right]$$

The learning follows an iterative procedure wherein the discriminator and generator are alternatively updated. Given a fixed $G$, the maximization subject to $D$ results in the optimal discriminator $D^\star(\mathbf{x}) = \frac{p_{\text{data}}(\mathbf{x})}{p_{\text{data}}(\mathbf{x}) + p_G(\mathbf{x})}$, whilst given this optimal $D^\star$, the minimization of $G$ turns into minimizing the Jensen-Shannon (JS) divergence between the data and model distributions: $D_{\text{JS}}\left(P_{\text{data}} \| P_G\right)$ [10]. At the Nash equilibrium of a game, the model distribution recovers the data distribution exactly: $P_G = P_{\text{data}}$, thus the discriminator $D$ now fails to differentiate real or fake data as $D(\mathbf{x}) = 0.5, \forall \mathbf{x}$.

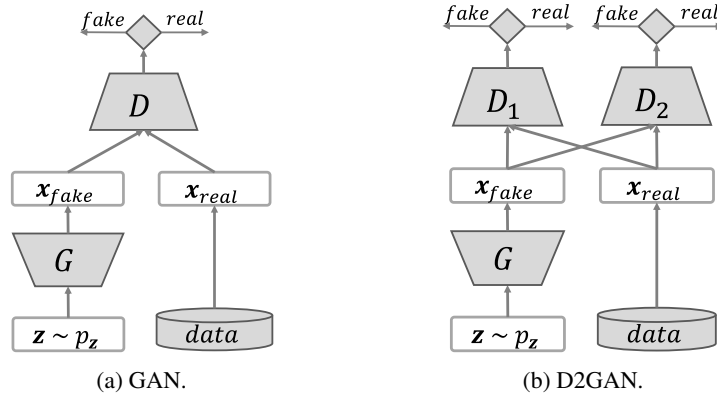

(a) GAN.                    (b) D2GAN.

Figure 1: An illustration of the standard GAN and our proposed D2GAN.

Since the JS divergence has been empirically proven to have the same nature as that of the reverse KL divergence [29, 15, 11], GAN suffers from the model collapsing problem, and thus its generated data samples have low level of diversity [20, 5].

## 3   Dual Discriminator Generative Adversarial Nets

To tackle GAN's problem of mode collapse, in what follows we present our main contribution of a framework that seeks an approximated distribution to effectively cover many modes of the multimodal data. Our intuition is based on GAN, but we formulate a three-player game that consists of two different discriminators $D_1$ and $D_2$, and one generator $G$. Given a sample $\mathbf{x}$ in data space, $D_1(\mathbf{x})$ rewards a high score if $\mathbf{x}$ is drawn from the data distribution $P_{\text{data}}$, and gives a low score if generated from the model distribution $P_G$. In contrast, $D_2(\mathbf{x})$ returns a high score for $\mathbf{x}$ generated from $P_G$ whilst giving a low score for a sample drawn from $P_{\text{data}}$. Unlike GAN, the scores returned by our discriminators are values in $\mathbb{R}^+$ rather than probabilities in $[0, 1]$. Our generator $G$ performs a similar role to that of GAN, i.e., producing data mapped from a noise space to synthesize the real data and then fool both two discriminators $D_1$ and $D_2$. All three players are parameterized by neural networks wherein $D_1$ and $D_2$ do not share their parameters. We term our proposed model *dual discriminator generative adversarial network* (D2GAN). Fig. 1b shows an illustration of D2GAN.

More formally, $D_1$, $D_2$ and $G$ now play the following three-player minimax optimization game:

$$\min_G \max_{D_1,D_2} \mathcal{J}(G,D_1,D_2) = \alpha \times \mathbb{E}_{\mathbf{x}\sim P_{\text{data}}}[\log D_1(\mathbf{x})] + \mathbb{E}_{\mathbf{z}\sim P_{\mathbf{z}}}[-D_1(G(\mathbf{z}))]$$

$$+ \mathbb{E}_{\mathbf{x}\sim P_{\text{data}}}[-D_2(\mathbf{x})] + \beta \times \mathbb{E}_{\mathbf{z}\sim P_{\mathbf{z}}}[\log D_2(G(\mathbf{z}))] \qquad (1)$$

wherein we have introduced hyperparameters $0 < \alpha, \beta \leq 1$ to serve two purposes. The first is to stabilize the learning of our model. As the output values of two discriminators are positive and unbounded, $D_1(G(\mathbf{z}))$ and $D_2(\mathbf{x})$ in Eq. (1) can become very large and have exponentially stronger impact on the optimization than $\log D_1(\mathbf{x})$ and $\log D_2(G(\mathbf{z}))$ do, rendering the learning unstable. To overcome this issue, we can decrease $\alpha$ and $\beta$, in effect making the optimization penalize $D_1(G(\mathbf{z}))$ and $D_2(\mathbf{x})$, thus helping to stabilize the learning. The second purpose of introducing $\alpha$ and $\beta$ is to control the effect of KL and reverse KL divergences on the optimization problem. This will be discussed in the following part once we have the derivation of our optimal solution.

Similar to GAN [10], our proposed network can be trained by alternatively updating $D_1$, $D_2$ and $G$. We refer to the supplementary material for the pseudo-code of learning parameters for D2GAN.

### 3.1 Theoretical analysis

We now provide formal theoretical analysis of our proposed model, that essentially shows that, given $G$, $D_1$ and $D_2$ are of enough capacity, i.e., in the nonparametric limit, at the optimal points, $G$ can recover the data distributions by minimizing both KL and reverse KL divergences between model and data distributions. We first consider the optimization problem with respect to (w.r.t) discriminators given a fixed generator.

**Proposition 1.** *Given a fixed $G$, maximizing $\mathcal{J}(G,D_1,D_2)$ yields to the following closed-form optimal discriminators $D_1^\star, D_2^\star$:*

$$D_1^\star(\mathbf{x}) = \frac{\alpha p_{\text{data}}(\mathbf{x})}{p_G(\mathbf{x})} \quad \text{and} \quad D_2^\star(\mathbf{x}) = \frac{\beta p_G(\mathbf{x})}{p_{\text{data}}(\mathbf{x})}$$

*Proof.* According to the induced measure theorem [12], two expectations are equal: $\mathbb{E}_{\mathbf{z}\sim P_{\mathbf{z}}}[f(G(\mathbf{z}))] = \mathbb{E}_{\mathbf{x}\sim P_G}[f(\mathbf{x})]$ where $f(\mathbf{x}) = -D_1(\mathbf{x})$ or $f(\mathbf{x}) = \log D_2(\mathbf{x})$. The objective function can be rewritten as below:

$$\mathcal{J}(G,D_1,D_2) = \alpha \times \mathbb{E}_{\mathbf{x}\sim P_{\text{data}}}[\log D_1(\mathbf{x})] + \mathbb{E}_{\mathbf{x}\sim P_G}[-D_1(\mathbf{x})]$$

$$+ \mathbb{E}_{\mathbf{x}\sim P_{\text{data}}}[-D_2(\mathbf{x})] + \beta \times \mathbb{E}_{\mathbf{x}\sim P_G}[\log D_2(\mathbf{x})]$$

$$= \int_{\mathbf{x}} [\alpha p_{\text{data}}(\mathbf{x})\log D_1(\mathbf{x}) - p_G D_1(\mathbf{x}) - p_{\text{data}}(\mathbf{x}) D_2(\mathbf{x}) + \beta p_G \log D_2(\mathbf{x})]\,\mathrm{d}\mathbf{x}$$

Considering the function inside the integral, given $\mathbf{x}$, we maximize this function w.r.t two variables $D_1, D_2$ to find $D_1^\star(\mathbf{x})$ and $D_2^\star(\mathbf{x})$. Setting the derivatives w.r.t $D_1$ and $D_2$ to 0, we gain:

$$\frac{\alpha p_{\text{data}}(\mathbf{x})}{D_1} - p_G(\mathbf{x}) = 0 \quad \text{and} \quad \frac{\beta p_G(\mathbf{x})}{D_2} - p_{\text{data}}(\mathbf{x}) = 0 \qquad (2)$$

The second derivatives: $-\alpha p_{\text{data}}(\mathbf{x})/D_1^2$ and $-\beta p_G(\mathbf{x})/D_2^2$ are non-positive, thus verifying that we have obtained the maximum solution and concluding the proof. $\square$

Next, we fix $D_1 = D_1^\star, D_2 = D_2^\star$ and find the optimal solution $G^\star$ for the generator $G$.

**Theorem 2.** *Given $D_1^\star, D_2^\star$, at the Nash equilibrium point $(G^\star, D_1^\star, D_2^\star)$ for minimax optimization problem of D2GAN, we have the following form for each component:*

$$\mathcal{J}(G^\star, D_1^\star, D_2^\star) = \alpha(\log\alpha - 1) + \beta(\log\beta - 1)$$

$$D_1^\star(\mathbf{x}) = \alpha \text{ and } D_2^\star(\mathbf{x}) = \beta, \forall \mathbf{x} \text{ at } p_{G^\star} = p_{data}$$

*Proof.* Substituting $D_1^\star, D_2^\star$ from Eq. (2) into the objective function in Eq. (1) of the minimax problem, we gain:

$$\mathcal{J}(G,D_1^\star,D_2^\star) = \alpha \times \mathbb{E}_{\mathbf{x}\sim P_{\text{data}}}\left[\log\alpha + \log\frac{p_{\text{data}}(\mathbf{x})}{p_G(\mathbf{x})}\right] - \alpha\int_{\mathbf{x}} p_G(\mathbf{x})\frac{p_{\text{data}}(\mathbf{x})}{p_G(\mathbf{x})}\,\mathrm{d}\mathbf{x}$$

$$- \beta\int_{\mathbf{x}} p_{\text{data}}\frac{p_G(\mathbf{x})}{p_{\text{data}}(\mathbf{x})}\,\mathrm{d}\mathbf{x} + \beta \times \mathbb{E}_{\mathbf{x}\sim P_G}\left[\log\beta + \log\frac{p_G(\mathbf{x})}{p_{\text{data}}(\mathbf{x})}\right]$$

$$= \alpha(\log\alpha - 1) + \beta(\log\beta - 1) + \alpha D_{\text{KL}}(P_{\text{data}}\|P_G) + \beta D_{\text{KL}}(P_G\|P_{\text{data}}) \qquad (3)$$

where $D_{\mathrm{KL}}\left(P_{\mathrm{data}}\|P_G\right)$ and $D_{\mathrm{KL}}\left(P_G\|P_{\mathrm{data}}\right)$ is the KL and reverse KL divergences between data and model (generator) distributions, respectively. These divergences are always nonnegative and only zero when two distributions are equal: $p_{G^\star} = p_{\mathrm{data}}$. In other words, the generator induces a distribution $p_{G^\star}$ that is identical to the data distribution $p_{\mathrm{data}}$, and two discriminators now fail to recognize the real or fake samples since they return the same score of 1 for both samples. This concludes the proof. $\qquad\square$

The loss of generator in Eq. (3) becomes an upper bound when the discriminators are not optimal. This loss shows that increasing $\alpha$ promotes the optimization towards minimizing the KL divergence $D_{\mathrm{KL}}\left(P_{\mathrm{data}}\|P_G\right)$, thus helping the generative distribution cover multiple modes, but may include potentially undesirable samples; whereas increasing $\beta$ encourages the minimization of the reverse KL divergence $D_{\mathrm{KL}}\left(P_G\|P_{\mathrm{data}}\right)$, hence enabling the generator capture a single mode better, but may miss many modes. By empirically adjusting these two hyperparameters, we can balance the effect of two divergences, and hence effectively avoid the mode collapsing issue.

## 3.2 Connection to f-GAN

Next we point out the relations between our proposed D2GAN and f-GAN – the model extends the Jensen-Shannon divergence (JSD) of GAN to more general divergences, specifically $f$-divergences [23]. A divergence in the $f$-divergence family has the following form:

$$D_f\left(P\|Q\right) = \int_{\mathcal{X}} q\left(x\right) f\left(\frac{q\left(x\right)}{p\left(x\right)}\right) \mathrm{d}x$$

where $f : \mathbb{R}_+ \to \mathbb{R}$ is a convex, lower-semicontinuous function satisfying $f\left(1\right) = 0$. This function has a convex conjugate function $f^*$, also known as Fenchel conjugate [13] : $f^*\left(t\right) = \sup_{u\in\mathrm{dom}_f}\left\{ut - f\left(u\right)\right\}$. The function $f^*$ is again convex and lower-semicontinuous.

Considering $P$ the true distribution and $Q$ the generator distribution, we resemble the learning problem in GAN by minimizing the $f$-divergence between $P$ and $Q$. Based on the variational lower bound of $f$-divergence proposed by Nguyen *et al.* [22], the objective function of f-GAN can be derived as follows:

$$\min_{\boldsymbol{\theta}} \max_{\boldsymbol{\phi}} F\left(\boldsymbol{\theta}, \boldsymbol{\phi}\right) = \mathbb{E}_{\mathbf{x}\sim P}\left[g_f\left(V_{\boldsymbol{\phi}}\left(\mathbf{x}\right)\right)\right] + \mathbb{E}_{\mathbf{x}\sim Q_{\boldsymbol{\theta}}}\left[-f^*\left(g_f\left(V_{\boldsymbol{\phi}}\left(\mathbf{x}\right)\right)\right)\right]$$

where $Q$ is parameterized by $\boldsymbol{\theta}$ (as the generator in GAN), $V_{\boldsymbol{\phi}} : \mathcal{X} \to \mathbb{R}$ is a function parameterized by $\boldsymbol{\phi}$ (as the discriminator in GAN) and $g_f : \mathbb{R} \to \mathrm{dom}_{f^*}$ is an output activation function (i.e., the discriminator's decision function) specific to the $f$-divergence used. Using appropriate functions $g_f$ and $f^*$ (see Tab. 2 in [23]), we recover the minimization of corresponding divergences such as JSD in GAN, KL (associated with discriminator $D_1$) and reverse KL (associated with discriminator $D_2$) of our D2GAN.

The f-GAN, however, only considers a single divergence. On the other hand, our proposed method combines KL and reserve KL divergences. Our idea is conceived upon pondering the advantages and disadvantages of these two divergences in covering multiple modes of data. Combining them into a unified objective function as in Eq. (3) helps us reversely engineer to finally obtain the optimization game in Eq. (1) that can be efficiently formulated and solved using the principle of GAN.

## 4 Experiments

In this section, we conduct comprehensive experiments to demonstrate the capability of improving mode coverage and the scalability of our proposed model on large-scale datasets. We use a synthetic 2D dataset for both visual and numerical verification, and four datasets of increasing diversity and size for numerical verification. We have made our best effort to compare the results of our method with those of the latest state-of-the-art GAN's variants by replicating experimental settings in the original work whenever possible.

For each experiment, we refer to the supplementary material for model architectures and additional results. Common points are: i) discriminators' outputs with *softplus* activations : $f\left(x\right) = \ln\left(1 + e^x\right)$, i.e., positive version of ReLU; (ii) Adam optimizer [16] with learning rate 0.0002 and the first-order momentum 0.5; (iii) minibatch size of 64 samples for training both generator and discriminators; (iv) Leaky ReLU with the slope of 0.2; and (v) weights initialized from an isotropic Gaussian: $\mathcal{N}\left(0, 0.01\right)$

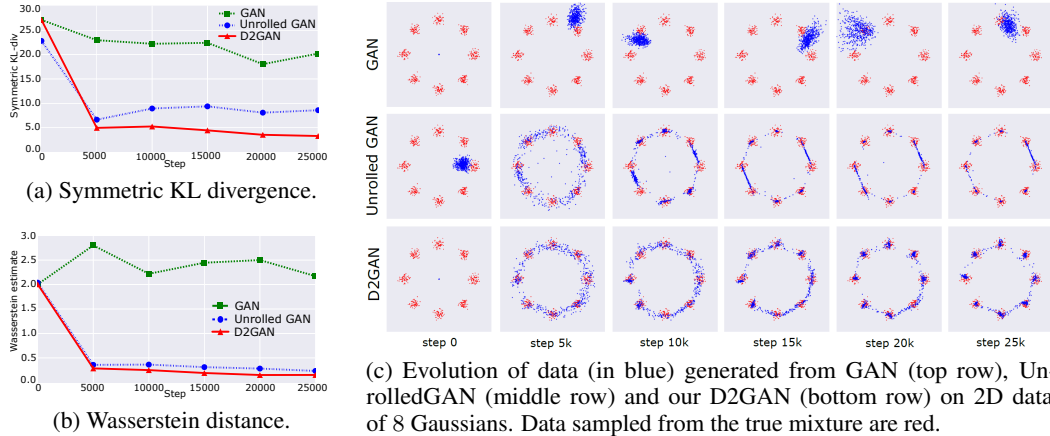

(a) Symmetric KL divergence.

(b) Wasserstein distance.

(c) Evolution of data (in blue) generated from GAN (top row), UnrolledGAN (middle row) and our D2GAN (bottom row) on 2D data of 8 Gaussians. Data sampled from the true mixture are red.

Figure 2: The comparison of standard GAN, UnrolledGAN and our D2GAN on 2D synthetic dataset.

and zero biases. Our implementation is in TensorFlow [1] and we have published a version for reference[1]. We now present our experiments on synthetic data followed by those on large-scale real-world datasets.

## 4.1 Synthetic data

In the first experiment, we reuse the experimental design proposed in [20] to investigate how well our D2GAN can deal with multiple modes in the data. More specifically, we sample training data from a 2D mixture of 8 Gaussian distributions with a covariance matrix $0.02I$ and means arranged in a circle of zero centroid and radius 2.0. Data in these low variance mixture components are separated by an area of very low density. The aim is to examine properties such as low probability regions and low separation of modes.

We use a simple architecture of a generator with two fully connected hidden layers and discriminators with one hidden layer of ReLU activations. This setting is identical, thus ensures a fair comparison with UnrolledGAN[2] [20]. Fig. 2c shows the evolution of 512 samples generated by our models and baselines through time. It can be seen that the regular GAN generates data collapsing into a *single* mode hovering around the valid modes of data distribution, thus reflecting the mode collapse in GAN. At the same time, UnrolledGAN and D2GAN distribute data around *all* 8 mixture components, and hence demonstrating the abilities to successfully learn multimodal data in this case. At the last steps, our D2GAN captures data modes more precisely than UnrolledGAN as, in each mode, the UnrolledGAN generates data that concentrate only on *several* points around the mode's centroid, thus seems to produce fewer samples than D2GAN whose samples fairly spread out the *entire* mode.

Next we further quantitatively compare the quality of generated data. Since we know the true distribution $p_{\text{data}}$ in this case, we employ two measures, namely symmetric KL divergence and Wasserstein distance. These measures compute the distance between the normalized histograms of 10,000 points generated from our D2GAN, UnrolledGAN and GAN to true $p_{\text{data}}$. Figs. 2a and 2b again clearly demonstrate the superiority of our approach over GAN and UnrolledGAN w.r.t both distances (lower is better); notably with Wasserstein metric, the distance from ours to the true distribution almost reduces to zero. These figures also demonstrate the stability of our D2GAN (red curves) during training as it is much less fluctuating compared with GAN (green curves) and UnrolledGAN (blue curves).

## 4.2 Real-world datasets

We now examine the performance of our proposed method on real-world datasets with increasing diversities and sizes. For networks containing convolutional layers, we closely follow the DCGAN's design [24]. We use strided convolutions for discriminators and fractional-strided convolutions for generator instead of pooling layers. Batch normalization is applied for each layer, except the

generator output layer and the discriminator input layers. We also use Leaky ReLU activations for discriminators, and use ReLU for generator, except its output is *tanh* since we rescale the pixel intensities into the range of [-1, 1] before feeding images to our model. Only one difference is that, for our model, initializing the weights from $\mathcal{N}(0, 0.01)$ yields slightly better results than from $\mathcal{N}(0, 0.02)$. We again refer to the supplementary material for detailed architectures.

### 4.2.1 Evaluation protocol

Evaluating the quality of image produced by generative models is a notoriously challenging due to the variety of probability criteria and the lack of a perceptually meaningful image similarity metric [29]. Even when a model can generate plausible images, it is not useful if those images are visually similar. Therefore, in order to quantify the performance of covering data modes as well as producing high quality samples, we use several different ad-hoc metrics for different experiments to compare with other baselines.

First we adopt the *Inception score* proposed in [27], which are computed by: $\exp\left(\mathbb{E}_\mathbf{x}\left[D_{\mathrm{KL}}\left(p\left(\mathrm{y}\mid\mathbf{x}\right)\parallel p\left(\mathrm{y}\right)\right)\right]\right)$, where $p\left(\mathrm{y}\mid\mathbf{x}\right)$ is the conditional label distribution for image $\mathbf{x}$ estimated using a pretrained Inception model [28], and $p\left(\mathrm{y}\right)$ is the marginal distribution: $p\left(\mathrm{y}\right) \approx {}^{1}\!/_{\mathrm{N}}\sum_{n=1}^{\mathrm{N}}p\left(\mathrm{y}\mid\mathbf{x}_n = G\left(\mathbf{z}_n\right)\right)$. This metric rewards good and varied samples, but sometimes is easily fooled by a model that collapses and generates to a very low quality image, thus fails to measure whether a model has been trapped into one bad mode. To address this problem, for labeled datasets, we further recruit the so-called MODE score introduced in [5]:

$$\exp\left(\mathbb{E}_\mathbf{x}\left[D_{\mathrm{KL}}\left(p\left(\mathrm{y}\mid\mathbf{x}\right)\parallel\tilde{p}\left(\mathrm{y}\right)\right)\right] - D_{\mathrm{KL}}\left(p\left(\mathrm{y}\right)\parallel\tilde{p}\left(\mathrm{y}\right)\right)\right)$$

where $\tilde{p}\left(\mathrm{y}\right)$ is the empirical distribution of labels estimated from training data. The score can adequately reflect the variety and visual quality of images, which is discussed in [5].

### 4.2.2 Handwritten digit images

We start with the handwritten digit images – MNIST [19] that consists of 60,000 training and 10,000 testing 28×28 grayscale images of digits from 0 to 9. Following the setting in [5], we first assume that the MNIST has 10 modes, representing connected component in the data manifold, associated with 10 digit classes. We then also perform an extensive grid search of different hyperparameter configurations, wherein our two regularized constants $\alpha, \beta$ in Eq. (1) are varied in {0.01, 0.05, 0.1, 0.2}. For a fair comparison, we use the same parameter ranges and fully connected layers for our network (c.f. the supplementary material for more details), and adopt results of GAN and mode regularized GAN (Reg-GAN) from [5].

For evaluation, we first train a simple, yet effective 3-layer convolutional nets[3] that can obtain 0.65% error on MNIST testing set, and then employ it to predict the label probabilities and compute MODE scores for generated samples. Fig. 3 (left) shows the distributions of MODE scores obtained by three models. Clearly, our proposed D2GAN significantly outperforms the standard GAN and Reg-GAN by achieving scores mostly around the maximum [8.0-9.0]. It is worthy to note that we did not observe substantial differences in the average MODE scores obtained by varying the network size through the parameter searching. We here report the result of the minimal network with the smallest number of layers and hidden units.

To study the effect of $\alpha$ and $\beta$, we inspect the results obtained by this minimal network with varied $\alpha, \beta$ in Fig. 3 (right). There is a pattern that, given a fixed $\alpha$, our D2GAN obtains better MODE score when increasing $\beta$ to a certain value, after which the score could significantly decrease.

**MNIST-1K.** The standard MNIST data with 10-mode assumption seems to be fairly trivial. Hence, based on this data, we test our proposed model on a more challenging one. We continue following the technique used in [5, 20] to construct a new 1000-class MNIST dataset (MNIST-1K) by stacking three randomly selected digits to form an RGB image with a different digit image in each channel. The resulting data can be assumed to contain 1,000 distinct modes, corresponding to the combinations of digits in 3 channels from 000 to 999.

In this experiment, we use a more powerful model with convolutional layers for discriminators and transposed convolutions for the generator. We measure the performance by the number of modes

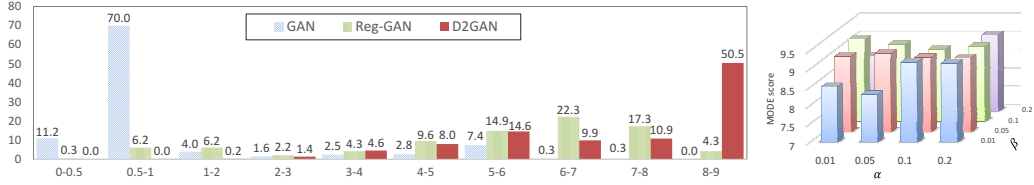

Figure 3: Distributions of MODE scores (left) and average MODE scores (right) with varied $\alpha$, $\beta$.

for which the model generated at least one in total 25,600 samples, and the reverse KL divergence between the model distribution (i.e., the label distribution predicted by the pretrained MNIST classifier used in the previous experiment) and the expected data distribution. Tab. 1 reports the results of our D2GAN compared with those of GAN, UnrolledGAN taken from [20], DCGAN and Reg-GAN from [5]. Our proposed method again clearly demonstrates the superiority over baselines by covering all modes and achieving the best distance that is close to zero.

Table 1: Numbers of modes covered and reverse KL divergence between model and data distributions.

| Model | GAN [20] | UnrolledGAN [20] | DCGAN [5] | Reg-GAN [5] | D2GAN |
|---|---|---|---|---|---|
| # modes covered | 628.0±140.9 | 817.4±37.9 | 849.6±62.7 | 955.5±18.7 | **1000.0±0.00** |
| $D_{\mathrm{KL}}$ (model$\|$ data) | 2.58±0.75 | 1.43±0.12 | 0.73±0.09 | 0.64±0.05 | **0.08±0.01** |

### 4.2.3 Natural scene images

We now extend our experiments to investigate the scalability of our proposed method on much more challenging large-scale image databases from natural scenes. We use three widely-adopted datasets: CIFAR-10 [17], STL-10 [6] and ImageNet [26]. CIFAR-10 is a well-studied dataset of 50,000 $32\times32$ training images of 10 classes: airplane, automobile, bird, cat, deer, dog, frog, horse, ship, and truck. STL-10, a subset of ImageNet, contains about 100,000 unlabeled $96\times96$ images, which is more diverse than CIFAR-10, but less so than the full ImageNet. We rescale all images down 3 times and train our networks on $32\times32$ resolution. ImageNet is a very large database of about 1.2 million natural images from 1,000 classes, normally used as the most challenging benchmark to validate the scalability of deep models. We follow the preprocessing in [18], except subsampling to $32\times32$ resolution. We use the code provided in [27] to compute the Inception score for 10 independent partitions of 50,000 generated samples.

Table 2: Inception scores on CIFAR-10.

| Model | Score |
|---|---|
| Real data | 11.24±0.16 |
| WGAN [2] | 3.82±0.06 |
| MIX+WGAN [3] | 4.04±0.07 |
| Improved-GAN [27] | 4.36±0.04 |
| ALI [8] | 5.34±0.05 |
| BEGAN [4] | 5.62 |
| MAGAN [30] | 5.67 |
| DCGAN [24] | 6.40±0.05 |
| DFM [31] | 7.72±0.13 |
| **D2GAN** | **7.15±0.07** |

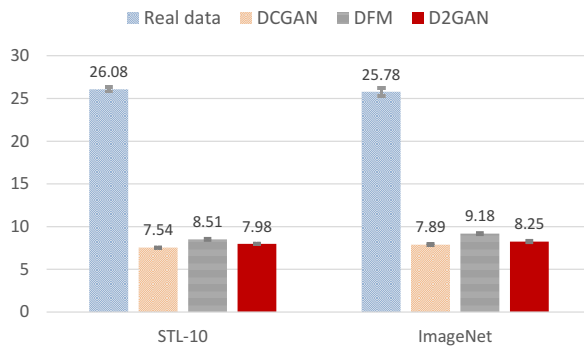

Figure 4: Inception scores on STL-10 and ImageNet.

Tab. 2 and Fig. 4 show the Inception scores on CIFAR-10, STL-10 and ImageNet datasets obtained by our model and baselines collected from recent work in literature. It is worthy to note that we only compare with methods trained in a completely unsupervised manner without label information. As the result, there exist 8 baselines on CIFAR-10 whilst only DCGAN [24] and denoising feature matching (DFM) [31] are available on STL-10 and ImageNet. We use our own TensorFlow implementation of DCGAN with the same network architecture with our model for fair comparisons. In all 3 experiments, the D2GAN fails to beat the DFM, but outperforms other baselines by large margins. The lower results compared with DFM suggest that using autoencoders for matching high-level features appears

to be an effective way to encourage the diversity. This technique is compatible with our method, thus integrating it could be a promising avenue for our future work.

Two discriminators $D_1$ and $D_2$ have almost identical architectures, thus they potentially can share parameters in many different schemes. We explore this direction by creating two version of our D2GAN with the same hyperparameter setting. The first version shares all parameters of $D_1$ and $D_2$ except the last (output) layer. This model has failed because the discriminator now contains much fewer parameters, rendering it unable to capture two inverse ratios of two density functions. The second one shares all parameters of $D_1$ and $D_2$ except the last two layers. This version performed better than the previous one, and could obtain promising Inception scores (7.01 on CIFAR10, 7.44 on STL10 and 7.81 on ImageNet), but these results are still worse than those of our proposed model without sharing parameters.

Finally, we show several samples generated by our proposed model trained on these three datasets in Fig. 5. Samples are fair random draws, not cherry-picked. It can be seen that our D2GAN is able to produce visually recognizable images of cars, trucks, boats, horses on CIFAR-10. The objects are getting harder to recognize, but the shapes of airplanes, cars, trucks and animals still can be identified on STL-10, and images with various backgrounds such as sky, underwater, mountain, forest are shown on ImageNet. This confirms the diversity of samples generated by our model.

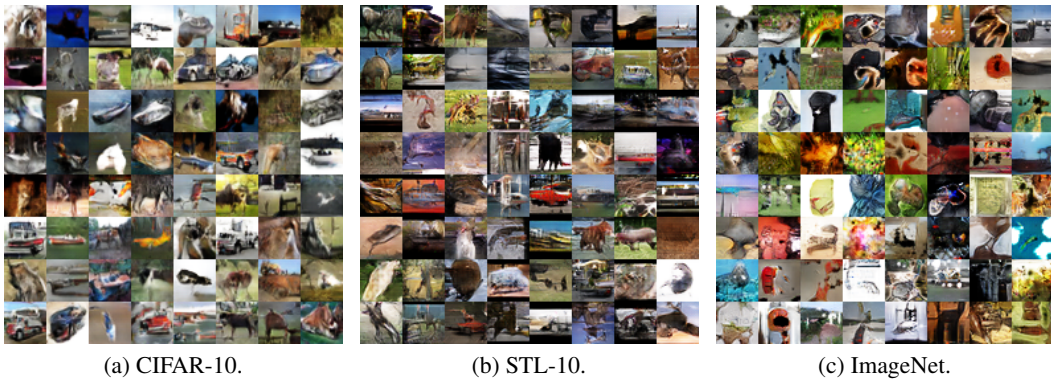

(a) CIFAR-10.  (b) STL-10.  (c) ImageNet.

Figure 5: Samples generated by our proposed D2GAN trained on natural image datasets. Due to the space limit, please refer to the supplementary material for larger plot.

## 5   Conclusion

To summarize, we have introduced a novel approach to combine Kullback-Leibler (KL) and reverse KL divergences in a unified objective function of the density estimation problem. Our idea is to exploit the complementary statistical properties of two divergences to improve both the quality and diversity of samples generated from the estimator. To that end, we propose a novel framework based on generative adversarial nets (GANs), which formulates a minimax game of three players: two discriminators and one generator, thus termed *dual discriminator GAN* (D2GAN). Given two discriminators fixed, the learning of generator moves towards optimizing both KL and reverse KL divergences simultaneously, and thus can help avoid mode collapse, a notorious drawback of GANs.

We have established extensive experiments to demonstrate the effectiveness and scalability of our proposed approach using synthetic and large-scale real-world datasets. Compared with the latest state-of-the-art baselines, our model is more scalable, can be trained on the large-scale ImageNet dataset, and obtains Inception scores lower than those of the combination of denoising autoencoder and GAN (DFM), but significantly higher than the others. Finally, we note that our method is orthogonal and could integrate techniques in those baselines such as semi-supervised learning [27], conditional architectures [21, 7, 25] and autoencoder [5, 31].

**Acknowledgments.** This work was partially supported by the Australian Research Council (ARC) Discovery Grant Project DP160109394.

## Footnotes

[1]https://github.com/tund/D2GAN

[2]We obtain the code of UnrolledGAN for 2D data from the link authors provided in [20].

[3]Network architecture is similar to https://github.com/fchollet/keras/blob/master/examples/mnist_cnn.py.

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
