[Supplementary Material]

# Supplementary Material for
## *"Dual Discriminator Generative Adversarial Nets"*

**Tu Dinh Nguyen, Trung Le, Hung Vu, Dinh Phung**
Deakin University, Geelong, Australia
Centre for Pattern Recognition and Data Analytics
{tu.nguyen, trung.l, hungv, dinh.phung}@deakin.edu.au

This document presents supplementary material to complement the manuscript entitled "*Dual Discriminator Generative Adversarial Nets*". We first describe the pseudo-code of learning algorithm for our proposed model, then present network architectures and hyperparameter settings in details and additional experimental results.

## 1   Framework

In our proposed method, two discriminators $D_1$ and $D_2$, and a generator $G$ are deep (convolutional) neural networks parameterized by $\boldsymbol{\theta}_{D_1}, \boldsymbol{\theta}_{D_2}$ and $\boldsymbol{\theta}_G$ respectively. The pseudo-code of learning those parameters is describe in Alg. 1. The discriminators and generator are alternatively updated using stochastic gradient ascent and descent, respectively. The update of $D_1, D_2$ can perform in parallel.

---
**Algorithm 1** Alternative training of D2GAN using stochastic gradient ascent and descent.

---
1: **for** number of training iterations **do**
2:    Sample a minibatch of M noise samples $\left(\mathbf{z}^{(1)}, \mathbf{z}^{(2)}, ..., \mathbf{z}^{(M)}\right)$ from the prior $p_{\mathbf{z}}$.
3:    Sample a minibatch of M data points $\left(\mathbf{x}^{(1)}, \mathbf{x}^{(2)}, ..., \mathbf{x}^{(M)}\right)$ from the data distribution $p_{\text{data}}$.

4:    Update the discriminator $D_1$ by ascending along its gradient:

$$\nabla_{\boldsymbol{\theta}_{D_1}} \frac{1}{M} \sum_{m=1}^{M} \left[ \alpha \times \log D_1\left(\mathbf{x}^{(m)}\right) - D_1\left(G\left(\mathbf{z}^{(m)}\right)\right) \right]$$

5:    Update the discriminator $D_2$ by ascending along its gradient:

$$\nabla_{\boldsymbol{\theta}_{D_2}} \frac{1}{M} \sum_{m=1}^{M} \left[ \beta \times \log D_2\left(G\left(\mathbf{z}^{(m)}\right)\right) - D_2\left(\mathbf{x}^{(m)}\right) \right]$$

6:    Sample a minibatch of M noise samples $\left(\mathbf{z}^{(1)}, \mathbf{z}^{(2)}, ..., \mathbf{z}^{(M)}\right)$ from the prior $p_{\mathbf{z}}$.
7:    Update the generator $G$ by descending along its gradient:

$$\nabla_{\boldsymbol{\theta}_G} \frac{1}{M} \sum_{m=1}^{M} \left[ \beta \times \log D_2\left(G\left(\mathbf{z}^{(m)}\right)\right) - D_1\left(G\left(\mathbf{z}^{(m)}\right)\right) \right]$$

8: **end for**

---

## 2   Details of the Experiments

In this section we present the network architectures, hyperparameter settings and additional experimental results of our proposed method for each experiment.

## 2.1 Synthetic 2D Gaussian data

First we describe the network architecture, hyperparameter settings and experimental details for experiment in Section 4.1. The true data is sampled from a 2D mixture of 8 Gaussian distributions with a covariance matrix $0.02I$ and means arranged in a circle of zero centroid and radius 2.0. We use a simple architecture of a generator with two fully connected hidden layers and discriminators with one hidden layer. All hidden layers contain the same number of 128 ReLU units. The input layer of generator contains 256 noise units sampled from isotropic multivariate Gaussian distribution $\mathcal{N}(0, I)$. We do not use batch normalization in any layer. We refer to Tab. 1 for more specifications of the network and hyperparameters.

Table 1: Network architecture and hyperparameters for 2D Gaussian data.

| Operation | Feature maps | Nonlinearity |
|---|---|---|
| $G(\mathbf{z}) : \mathbf{z} \sim \mathcal{N}(0, I)$ | 256 | |
| Fully connected | 128 | ReLU |
| Fully connected | 128 | ReLU |
| Fully connected | 2 | Linear |
| $D_1(\mathbf{x}), D_2(\mathbf{x})$ | 2 | |
| Fully connected | 128 | ReLU |
| Fully connected | 1 | Softplus |
| Learning rate | 0.0002 | |
| Batch size | 512 | |
| Number of iterations | 25,000 | |
| Leaky ReLU slope | 0.2 | |
| Regularization constants | $\alpha = 0.1, \beta = 1.0$ | |
| Optimizer | Adam($\beta_1 = 0.5, \beta_2 = 0.999$) | |
| Weight, bias initialization | $\mathcal{N}(\mu = 0, \sigma = 0.01), 0$ | |

Besides the results presented in the main manuscript, we have conducted additional experiments to investigate the performance of our proposed D2GAN on this synthetic data. First we examine the effect of individual discriminator by teasing out the two discriminators in the loss function. The model with only $D_1$ fails to learn the generator because $D_1(\mathbf{x})$ became very large while $D_1(G(\mathbf{z}))$ very small (since $D_1$ targets $P_{data}/P_G$), and thus $D_1$ easily dominated $G$. On the other hand, the model with only $D_2$ had learned a much better generator since $D_2(G(\mathbf{z}))$ was large, thus $G$ could compete with $D_2$ and generate samples to a certain extent of quality; nonetheless the performance was still far worse than the current model where both $D_1$ and $D_2$ employed.

We next setup two more experiments wherein the initialized distributions of generator and true data are disjoint. We freeze the generator and train two discriminators in the first experiment, whilst we update discriminators once after every 500 updates of generator in the second experiment. The learning behaviors are (as expected): in the first experiment, the loss would go to infinity since discriminators target the ratios of two densities whose denominators are close to zero. In the second experiment, the generator would put all mass to a single point since discriminators are not sufficiently trained to push generated samples closer to true data. As the result, jointly training and frequently updating generator and discriminators are important considerations for successful learning for our proposed model.

## 2.2 Handwritten digit images

Next we describe the convolutional neural network that is used to predict the label probabilities for MNIST images. The network contains 2 convolutional layers, followed by a max-pooling and two fully connected layers, with ReLU and Softmax activations. Dropout is used in the max-pooling layer and the fully connected layer next to it. This is a simple, yet effective model that can obtain 0.65% error on MNIST testing set. Tab. 2 reports the network design and hyperparameter settings, where we note that the dropout rate denotes the probability of dropping a neuron, thus 0.0 means no dropout.

Table 2: Network architecture and hyperparameters of convolutional net serving as a MNIST classifier.

| Operation | Kernel | Strides | Feature maps | Dropout | Nonlinearity |
|---|---|---|---|---|---|
| Input | | | $28 \times 28 \times 1$ | | |
| Convolution | $3 \times 3$ | $1 \times 1$ | 32 | 0.0 | ReLU |
| Convolution | $3 \times 3$ | $1 \times 1$ | 64 | 0.0 | ReLU |
| Max-pooling | $2 \times 2$ | $2 \times 2$ | 64 | 0.25 | |
| Fully connected | | | 128 | 0.5 | ReLU |
| Fully connected | | | 10 | 0.0 | Softmax |
| Batch size | 128 | | | | |
| Number of epochs | 30 | | | | |
| Leaky ReLU slope | 0.2 | | | | |
| Learning rate | 0.001 | | | | |
| Optimizer | Adam($\beta_1 = 0.9, \beta_2 = 0.999$) | | | | |
| Weight, bias initialization | $\mathcal{N}(\mu = 0, \sigma = 0.01), 0$ | | | | |

### 2.2.1 The standard MNIST

In the grid search on the standard MNIST dataset, we use fully connected layers and refer to Tab. 3 for parameter ranges of all networks. This is an extremely extensive searching that ran at full load on 10 GPU cards for about 10 days. We did not observe substantial differences in the average MODE scores obtained by different network sizes through the parameter searching. The result reported in the main manuscript is obtained by our minimal network with the smallest number of layers (2 for all generator and discriminators) and hidden units (256 for discriminators and 400 for generator) and no dropout.

Table 3: Grid search specification on the standard MNIST dataset.

| Setting | Description | GAN | Reg-GAN | D2GAN |
|---|---|---|---|---|
| nLayerG | number of layers in $G$ | {2, 3, 4, 5} | {2, 3, 4} | {2, 3, 4} |
| nLayerD | number of layers in $D$ | {2, 3, 4, 5} | {2, 3, 4} | {2, 3, 4} |
| sizeG | number of neurons in $G$ | {400, 800, 1600, 3200} | {400, 800, 1600, 3200} | {400, 800, 1600, 3200} |
| sizeD | number of neurons in $D$ | {128, 256, 512, 1024} | {256, 512, 1024} | {256, 512, 1024} |
| dropoutD | is to use dropout in $D$ | {True, False} | {True, False} | {True, False} |
| optimG | to use Adam or SGD for $G$ | {SGD, Adam} | {SGD, Adam} | Adam |
| optimD | to use Adam or SGD for $D$ | {SGD, Adam} | {SGD, Adam} | Adam |
| lr | learning rate | {0.01, 0.001, 0.0001} | {0.01, 0.001, 0.0001} | {0.0002, 0.0001} |
| $\alpha$ | regularization constant | – | – | {0.01, 0.05, 0.1, 0.2} |
| $\beta$ | regularization constant | – | – | {0.01, 0.05, 0.1, 0.2} |

### 2.2.2 MNIST-1K

In the experiment on the extended MNIST-1K dataset, we use a network with convolutional layers for discriminators and transposed convolutions for the generator, and employ batch normalization for several layers. This is a more powerful model than the one applied on the standard MNIST. We refer to Tab. 4 for network architecture and hyperparameter settings. BN is short for batch normalization. Fig. 1 shows image with RGB color channels generated by our D2GAN. It can be visually observed that our proposed model can generate very good quality and diverse images in each color channel. In addition, different numbers can be seen across 3 channels, thus demonstrating that our D2GAN is capable of generating samples of all 1,000 classes ranging from 000 to 999.

### 2.3 Natural scene datasets

In the last experiments on three large-scale natural scene datasets (CIFAR-10, STL-10, ImageNet), we closely follow the network architecture and training procedure of DCGAN. Tabs. (5, 6, 7) report the specifications of our models trained on CIFAR-10, STL-10 and Imagenet datasets, respectively. BN is short for batch normalization. The training is terminated after 250 epochs scanning through the entire training data. To inspect the progress in generating data of our model through the training, we randomly fix a set of noise samples, and then generate the corresponding data samples after some certain epochs. Fig. 2 shows the evolution of images generated by our proposed method for the three

Table 4: Network architecture and hyperparameters for MNIST-1K dataset.

| Operation | Kernel | Strides | Feature maps | BN? | Nonlinearity |
|---|---|---|---|---|---|
| $G(\mathbf{z}) : \mathbf{z} \sim \mathcal{N}(0, \boldsymbol{I})$ | | | 256 | | |
| Fully connected | | | 4×4×512 | $\sqrt{}$ | ReLU |
| Transposed convolution | 5×5 | 2×2 | 256 | $\sqrt{}$ | ReLU |
| Transposed convolution | 5×5 | 2×2 | 128 | $\sqrt{}$ | ReLU |
| Transposed convolution | 5×5 | 2×2 | 3 | × | Tanh |
| $D_1(\mathbf{x}), D_2(\mathbf{x})$ | | | 28×28×3 | | |
| Convolution | 5×5 | 2×2 | 128 | × | Leaky ReLU |
| Convolution | 5×5 | 2×2 | 256 | $\sqrt{}$ | Leaky ReLU |
| Convolution | 5×5 | 2×2 | 512 | $\sqrt{}$ | Leaky ReLU |
| Fully connected | | | 1 | × | Softplus |
| Batch size | 64 | | | | |
| Number of epochs | 30 | | | | |
| Leaky ReLU slope | 0.2 | | | | |
| Learning rate | 0.0002 | | | | |
| Regularization constants | $\alpha = 0.1, \beta = 0.1$ | | | | |
| Optimizer | Adam($\beta_1 = 0.5, \beta_2 = 0.999$) | | | | |
| Weight, bias initialization | $\mathcal{N}(\mu = 0, \sigma = 0.01), 0$ | | | | |

Figure 1: Images generated by D2GAN trained on MNIST-1K dataset.

datasets after 0, 50, 100, 200 and 250 epochs. It can be seen that the image quality is improved with images becoming sharper and objects more recognizable over time. Finally, Figs. (3, 4, 5) respectively are the enlarged versions of Figs. (5a, 5b, 5c) in the main manuscript. These images are generated by our D2GAN when the training completely finishes.

Table 5: Network architecture and hyperparameters for CIFAR-10 dataset.

| Operation | Kernel | Strides | Feature maps | BN? | Nonlinearity |
|---|---|---|---|---|---|
| $G\left(\mathbf{z}\right):\mathbf{z}\sim\text{Uniform}\left[-1,1\right]$ | | | 100 | | |
| Fully connected | | | 4×4×512 | $\checkmark$ | ReLU |
| Transposed convolution | 5×5 | 2×2 | 256 | $\checkmark$ | ReLU |
| Transposed convolution | 5×5 | 2×2 | 128 | $\checkmark$ | ReLU |
| Transposed convolution | 5×5 | 2×2 | 3 | $\times$ | Tanh |
| $D_1\left(\mathbf{x}\right),D_2\left(\mathbf{x}\right)$ | | | 32×32×3 | | |
| Convolution | 5×5 | 2×2 | 128 | $\times$ | Leaky ReLU |
| Convolution | 5×5 | 2×2 | 256 | $\checkmark$ | Leaky ReLU |
| Convolution | 5×5 | 2×2 | 512 | $\checkmark$ | Leaky ReLU |
| Fully connected | | | 1 | $\times$ | Softplus |
| Batch size | 64 | | | | |
| Number of iterations | 250 | | | | |
| Leaky ReLU slope | 0.2 | | | | |
| Learning rate | 0.0002 | | | | |
| Regularization constants | $\alpha=0.01,\ \beta=0.01$ | | | | |
| Optimizer | Adam($\beta_1=0.5,\beta_2=0.999$) | | | | |
| Weight, bias initialization | $\mathcal{N}\left(\mu=0,\sigma=0.01\right),0$ | | | | |

Table 6: Network architecture and hyperparameters for STL-10 dataset.

| Operation | Kernel | Strides | Feature maps | BN? | Nonlinearity |
|---|---|---|---|---|---|
| $G\left(\mathbf{z}\right):\mathbf{z}\sim\text{Uniform}\left[-1,1\right]$ | | | 100 | | |
| Fully connected | | | 4×4×512 | $\checkmark$ | ReLU |
| Transposed convolution | 5×5 | 2×2 | 256 | $\checkmark$ | ReLU |
| Transposed convolution | 5×5 | 2×2 | 128 | $\checkmark$ | ReLU |
| Transposed convolution | 5×5 | 2×2 | 3 | $\times$ | Tanh |
| $D_1\left(\mathbf{x}\right),D_2\left(\mathbf{x}\right)$ | | | 32×32×3 | | |
| Convolution | 5×5 | 2×2 | 128 | $\times$ | Leaky ReLU |
| Convolution | 5×5 | 2×2 | 256 | $\checkmark$ | Leaky ReLU |
| Convolution | 5×5 | 2×2 | 512 | $\checkmark$ | Leaky ReLU |
| Fully connected | | | 1 | $\times$ | Softplus |
| Batch size | 64 | | | | |
| Number of iterations | 250 | | | | |
| Leaky ReLU slope | 0.2 | | | | |
| Learning rate | 0.0001 | | | | |
| Regularization constants | $\alpha=0.1,\ \beta=0.1$ | | | | |
| Optimizer | Adam($\beta_1=0.5,\beta_2=0.999$) | | | | |
| Weight, bias initialization | $\mathcal{N}\left(\mu=0,\sigma=0.01\right),0$ | | | | |

Table 7: Network architecture and hyperparameters for ImageNet dataset.

| Operation | Kernel | Strides | Feature maps | BN? | Nonlinearity |
|---|---|---|---|---|---|
| $G(\mathbf{z}) : \mathbf{z} \sim \mathcal{N}(0, \boldsymbol{I})$ | | | 100 | | |
| Fully connected | | | 4×4×512 | √ | ReLU |
| Transposed convolution | 5×5 | 2×2 | 256 | √ | ReLU |
| Transposed convolution | 5×5 | 2×2 | 128 | √ | ReLU |
| Transposed convolution | 5×5 | 2×2 | 3 | × | Tanh |
| $D_1(\mathbf{x}), D_2(\mathbf{x})$ | | | 32×32×3 | | |
| Convolution | 5×5 | 2×2 | 128 | × | Leaky ReLU |
| Convolution | 5×5 | 2×2 | 256 | √ | Leaky ReLU |
| Convolution | 5×5 | 2×2 | 512 | √ | Leaky ReLU |
| Fully connected | | | 1 | × | Softplus |
| Batch size | 64 | | | | |
| Number of iterations | 250 | | | | |
| Leaky ReLU slope | 0.2 | | | | |
| Learning rate | 0.0002 | | | | |
| Regularization constants | $\alpha = 0.0001, \beta = 0.01$ | | | | |
| Optimizer | Adam($\beta_1 = 0.5, \beta_2 = 0.999$) | | | | |
| Weight, bias initialization | $\mathcal{N}(\mu = 0, \sigma = 0.01), 0$ | | | | |

Figure 2: Evolution of images generated by D2GAN trained on CIFAR-10 (top row), STL-10 (middle row) and ImageNet (bottom row) datasets.

Figure 3: Images generated by D2GAN trained on CIFAR-10 dataset.

Figure 4: Images generated by D2GAN trained on STL-10 dataset.

Figure 5: Images generated by D2GAN trained on ImageNet dataset.