[Reviews · NeurIPS 2017]

Reviewer 1



The paper proposes to train GANs by minimizing lower bounds of KL and reverse KL. The KL and the reverse KL costs are weighted by two hyperparameters. The KL and reverse KL estimators were previously mentioned in f-GAN. It seems that this paper rediscovers the estimators. The theoretical analysis is consistent with the f-GAN properties. Pros: - It is good to know that the minimization of the symmetric KL works well. The experiments show good results on 2D toy data, MNIST-1K and natural images. Cons: - It is not clear how the KL and reverse KL estimators were derived. f-GAN is not mentioned in that context. - It would be nice to see a discussion of the limitations. What will happen if the discriminators are trained till convergence? The loss would go to infinity if measuring KL for distributions with disjoint supports. What will happen if the discriminators are not updated frequently? The generator can put all mass to a single point. - It would be good to see whether D2GAN works OK without batch normalization in the generator. Questions: 1) Have you tried using just one discriminator network with 2 outputs to produce D1(x) and D2(x)? Was it working worse? 2) Were two discriminator networks used in the experiments? That would be a bit unfair to the baselines with just one discriminator net. Minor typos: - The $p_\mathrm{data}$ style is not consistent in the document. - Sometimes (x) is missing next to p_G(x) or p_data(x). Maybe it is intended. - The optimal discriminators in Theorem 2 should be: D1 = alpha, D2 = beta. Update: I have read the rebuttal and I thank the authors for the extra experiments. The authors should mention that the estimated KL and reverse KL are just lower bounds, if the discriminators are not optimal.

Reviewer 2



This paper presents a variant of generative adversarial networks (GANs) that utilizes two discriminators, one tries to assign high scores for data, and the other tries to assign high scores for the samples, both discriminating data from samples, and the generator tries to fool both discriminators. It has been shown in section 3 that the proposed approach effectively optimizes the sum of KL and reverse KL between generator distribution and data distribution in the idealized non-parametric setup, therefore encouraging more mode coverage than other GAN variants. The paper is quite well written and the formulation and analysis seems sound and straightforward. The proposed approach is evaluated on toy 2D points dataset as well as more realistic MNIST, CIFAR-10, STL and ImageNet datasets. I have one concern about the new formulation, as shown in Proposition 1, the optimal discriminators have the form of density ratios. Would this cause instability issues? The density ratios can vary wildly from 0 to infinity when the modes of the model and the data do not match well. On the other hand, it is very hard for neural nets to learn to predict extremely large values with finite weights. On the toy dataset, however, the proposed D2GAN seems to be more stable than standard GAN and unrolled GAN, is this in general true though? I appreciate the authors’ effort in doing an extensive hyperparameter search for both the proposed method and the baselines on the MNIST dataset. However, for a better comparison, it feels we should train the same generator network using all the methods compared, and tune the other hyperparameters for that network architecture (or a few networks). Also, learning rate seems to be the most important hyperparameter but it is not searched extensively enough. The paper claims they can scale up their proposed method to ImageNet, but I feel this is a little bit over-claiming, as they only tried on a subset of ImageNet and downsampled all images to 32x32, while there are already existing methods that can generate ImageNet samples for much higher resolution. Overall the paper seems like a valid addition to the GAN literature, but doesn’t seem to change state-of-the-art too much compared to other approaches.

Reviewer 3



UPDATE (after rebuttal): Thanks for your feedback. 1. Tease out D1, D2: I think it's good to include these nevertheless (at least in supplemental information) as it's an interesting piece of experimental evidence. 3. Share parameters D1 and D2: Please include these results in the final version as well. I have increased the score to 8. -------- The paper proposes dual discriminator GAN (D2GAN), which uses two discriminators and a different loss function for training the discriminators. The approach is clearly explained and I enjoyed reading the paper. The experiments on MoG, MNIST-1K, CIFAR-10, STL-10, ImageNet support the main claims. Overall, I think this is a good paper and I vote for acceptance. I have a few questions / suggestions for improvement: Eq (1) uses D1 and D2 as well as a new loss function for the discriminator. Have you tried using only D1 or D2? I think it'd be useful to report this baseline so that we can tease out the two effects. Interestingly, the loss function obtained by using only D1 in eq (1), has an interesting connection with the KL importance estimation procedure (KLIEP). See also eq (12) in the paper: Learning in Implicit Generative Models Shakir Mohamed, Balaji Lakshminarayanan https://arxiv.org/pdf/1610.03483.pdf IIUC, you trained two separate networks for the discriminators D1 and D2. Have you tried sharing the parameters and just modifying the loss function? This reduces the number of parameters and makes the method more appealing. In particular, D1 targets p_data/p_g and D2 targets p_g/p_data. Since they are targeting the inverses of each other, it might be useful to have one network to model the log ratio p_data/p_g and use a hybrid loss for the ratio. If two networks are really necessary, it'd be helpful to add some insight on why this is the case (is it numerical issue, does it help the optimization etc)